# PUCA: Patch-Unshuffle and Channel Attention for Enhanced Self-Supervised Image Denoising

**Hyemi Jang**[1], **Junsung Park**[1], **Dahuin Jung**[1], **Jaihyun Lew**[2], **Ho Bae**[3,*] **Sungroh Yoon**[1,2,*]

[1]Department of Electrical and Computer Engineering, Seoul National University
[2]Interdisciplinary Program in Artificial Intelligence, Seoul National University
[3]Department of Cyber Security, Ewha Womans University

wkdal9512@snu.ac.kr      jerryray@snu.ac.kr      annajung0625@snu.ac.kr
fudojhl@snu.ac.kr        hobae@ewha.ac.kr        sryoon@snu.ac.kr

## Abstract

Although supervised image denoising networks have shown remarkable performance on synthesized noisy images, they often fail in practice due to the difference between real and synthesized noise. Since clean-noisy image pairs from the real world are extremely costly to gather, self-supervised learning, which utilizes noisy input itself as a target, has been studied. To prevent a self-supervised denoising model from learning identical mapping, each output pixel should not be influenced by its corresponding input pixel; This requirement is known as $\mathcal{J}$-invariance. Blind-spot networks (BSNs) have been a prevalent choice to ensure $\mathcal{J}$-invariance in self-supervised image denoising. However, constructing variations of BSNs by injecting additional operations such as downsampling can expose blinded information, thereby violating $\mathcal{J}$-invariance. Consequently, convolutions designed specifically for BSNs have been allowed only, limiting architectural flexibility. To overcome this limitation, we propose PUCA, a novel $\mathcal{J}$-invariant U-Net architecture, for self-supervised denoising. PUCA leverages patch-unshuffle/shuffle to dramatically expand receptive fields while maintaining $\mathcal{J}$-invariance and dilated attention blocks (DABs) for global context incorporation. Experimental results demonstrate that PUCA achieves state-of-the-art performance, outperforming existing methods in self-supervised image denoising.

## 1 Introduction

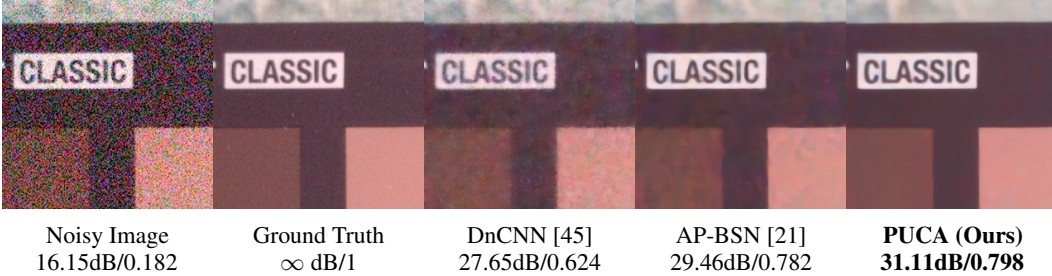

| Noisy Image | Ground Truth | DnCNN [45] | AP-BSN [21] | **PUCA (Ours)** |
|:---:|:---:|:---:|:---:|:---:|
| 16.15dB/0.182 | $\infty$ dB/1 | 27.65dB/0.624 | 29.46dB/0.782 | **31.11dB/0.798** |

Figure 1: **Visual comparison with other denoising methods on the SIDD validation dataset [1].** DnCNN was trained with real clean-noisy pairs of SIDD. AP-BSN [21] and PUCA were trained in a self-supervised manner solely on noisy images.

---

*Corresponding Authors

37th Conference on Neural Information Processing Systems (NeurIPS 2023).

Image denoising is a traditional task in computer vision, aiming to recover a clean image from a noisy one. With the advent of convolutional neural networks (CNNs) [17], deep learning-based methods have dominated the field. [23, 24, 46, 30, 47] These methods train a denoiser in a supervised manner using synthesized clean-noisy image pairs. However, denoising models trained on synthesized image pairs have limitations when dealing with real noise because of the domain gap between synthetic and real noise [47]. To overcome the limitations, several researchers [1, 5] have collected real-world clean-noisy image pairs in a controlled environment. However, this process requires a lot of manual effort and post-processing, and it is challenging to collect enough data, resulting in limited generalization ability. Therefore, self-supervised denoising methods have been proposed to train a denoising model solely relying on noisy images.

In self-supervised learning, where the model is trained using an identical image as input and target, it is essential to satisfy $\mathcal{J}$-invariance [4] to prevent the model from learning an identity mapping. $\mathcal{J}$-invariance refers to eliminating the influence of each pixel on the corresponding output pixel. Blind-spot networks (BSN), which basically meet the $\mathcal{J}$-invariance requirement, have been introduced to effectively denoise noisy inputs in a self-supervised manner [18, 36, 21]. BSNs [36, 21] achieve the requirement by incorporating centrally masked convolution, where the central pixel of the kernel has a zero value, and dilated convolutions. BSNs can be constructed with two choices of layers only, centrally masked convolutions and dilated convolutions, and cannot utilize varying operations such as downsampling. This constraint arises from the necessity to maintain $\mathcal{J}$-invariance. Consequently, the previously proposed architectures have been restricted due to the harsh requirement. In contrast, supervised denoising models enjoy more flexibility in terms of network structures since real clean-noisy image pairs are provided, alleviating the need for $\mathcal{J}$-invariance. This flexibility enabled the exploration of diverse architectures, and recently, the encoder-decoder based U-Net architectures [2, 8, 19, 33, 41, 43, 44], which preserve hierarchical multi-scale representations, have become prevalent.

As highlighted by the success of supervised denoising models, aggregating spatial information plays a critical role in effective denoising. In particular, U-Net architectures have shown superior improvements in modeling long-range dependencies and coarse-to-fine representation [2, 8, 19, 33, 41, 43, 44]. To benefit from the U-Net structure [27] in self-supervised learning, we propose **patch-unshuffle/shuffle**, a novel downsampling/upsampling technique that preserves $\mathcal{J}$-invariance. By incorporating downsampling/upsampling as a design option for BSN, we increase flexibility in network structures. Furthermore, patch-unshuffle expands receptive fields and utilizes multi-scale representation in BSN. We also introduce the **dilated attention block (DAB)**, a $\mathcal{J}$-invariant channel attention mechanism incorporating global information. DABs effectively remove noise by attentive connections between instances sub-sampled via patch-unshuffle. Hence, we integrate **Patch-Unshuffle and Channel Attention (PUCA)**, a novel U-Net architecture, while meeting the $\mathcal{J}$-invariant requirement. PUCA significantly outperforms existing state-of-the-art self-supervised denoising methods and unsupervised/unpaired approaches. As shown in Figure 1, PUCA shows comparable denoising quality with the ground truth. Our contributions can be summarized as follows:

- We propose patch-unshuffle/shuffle, a $\mathcal{J}$-invariant downsampling/upsampling method that effectively utilizes multi-scale representation and expands receptive fields. Moreover, patch-unshuffle/shuffle unleashes the constrained architecture design for blind-spot networks (BSNs).

- We introduce dilated attention blocks (DABs), which effectively incorporate global context through channel attention. The combination of DAB with patch-unshuffle/shuffle provides an advantage for noise removal by leveraging information from an ensemble of subsamples.

- By taking advantage of patch-unshuffle/shuffle and dilated attention block (DAB), PUCA, our proposed $\mathcal{J}$-invariant U-Net, outperforms other self-supervised and unpaired solutions by a substantial margin. Code is available at `https://github.com/HyemiEsme/PUCA`

## 2 Preliminaries

### 2.1 $\mathcal{J}$-invariant Network

In self-supervised learning, the denoising model needs to maintain $\mathcal{J}$-invariance to prevent learning identity mapping. $\mathcal{J}$-invariance is to remove the influence of the input pixel on the corresponding



Figure 2: **Dependency between the input and output pixels with a centrally masked convolution and d-dilated convolutions.** The green pixels indicate the pixels that are dependent on the center pixel in input, while the blue pixels represent the area independent of the central pixel. The yellow pixels represent the convolution weights.

output pixel. Since spatially correlated noise can expose evidence for the masked pixels, previous works [18, 4, 36] assume zero-mean pixel-wise independent noise. We also adopt the same noise assumption, and the definition of $\mathcal{J}$-invariance introduced in Noise2Self [4] is as follows.

**Definition 1** (Batson and Royer, 2019) Consider a partition $\mathcal{J}$ of the dimensions $\{1, ..., m\}$, an observed noisy signal $x$, and a sub-sample $x_J$ of $x$ limited to $J \in \mathcal{J}$. Let $g : \mathbb{R}^m \to \mathbb{R}^m$ be a function. A function $g$ is $J$-invariant if the value of $g(x)_J$ does not depend on the particular value of $x_J$; $g$ is $\mathcal{J}$-invariant if it is $J$-invariant for every $J \in \mathcal{J}$.

The $\mathcal{J}$-invariant network is trained to predict the value of each pixel using that of its surrounding pixels. To demonstrate the correspondence of self-supervised loss for supervised loss, we adopt the proposition from $\mathcal{J}$-invariant definition of Noise2Self [4].

**Proposition 1** (Batson and Royer, 2019) Let $x$ (a noisy image) be an unbiased estimator of $y$ (a clean image), denoted as $\mathbb{E}[x|y] = y$. Considering the $\mathcal{J}$-invariant function $g$, then the self-supervised loss,

$$\mathbb{E}||g(x) - x||^2 = \mathbb{E}||g(x) - y||^2 + \mathbb{E}||x - y||^2 \tag{1}$$

Under the assumption of zero-mean pixel-wise independent noise, the self-supervised loss is equal to the general supervised loss in addition to the variance of the noise. Therefore, by training the $\mathcal{J}$-invariant function $g$ using the self-supervised loss, $g$ can learn a way of eliminating the encoded general noise.

**Blind-spot network (BSN)**  BSNs [18, 20, 36] incorporate blind spots to the denoising framework to preserve $\mathcal{J}$-invariance in self-supervised learning. D-BSN [36] is a popular BSN architecture that utilizes centrally masked convolution in the first layer to exclude input pixel information. To maintain $\mathcal{J}$-invariance, D-BSN employs dilated convolution with dilation 2 when using a $3 \times 3$ centrally masked convolution and dilation 3 when using a $5 \times 5$ centrally masked convolution. The dependence of output pixels on the central pixel of input after stacking centrally masked convolution and dilated convolution is illustrated in Figure 2. The combination of the centrally masked convolution and the dilated convolution satisfies the $\mathcal{J}$-invariance.

### 2.2  Pixel-Shuffle Downsampling

Contrary to the assumption of zero-mean pixel-wise independent noise in BSN, real noise demonstrates correlations that enable noise prediction from neighboring pixels. To address this, Zhou et al. [47] introduced pixel-shuffle downsampling (PD), allowing denoisers trained on synthetic noise to handle real noise effectively by breaking spatial noise correlation. However, the analysis of AP-BSN [21] on real noise in SIDD reveals that images downsampled by the PD with a small stride factor still exhibit spatial noise correlation. To mitigate this, PD with a large stride factor is used during training. Nonetheless, using a PD with a large stride factor leads to a loss of input details. To preserve the details, a PD with a small stride factor is employed during testing. We also adopt asynchronous PD, utilizing different PD sizes for train and test phases.

## 3  Method

We propose PUCA, a novel $\mathcal{J}$-invariant U-Net as shown in Figure 3. Initially, we provide an overview of PUCA. Subsequently, we delve into the two essential elements of PUCA: (1) Patch Unshuffle/Shuffle and (2) Dilated Attention Block (DAB).

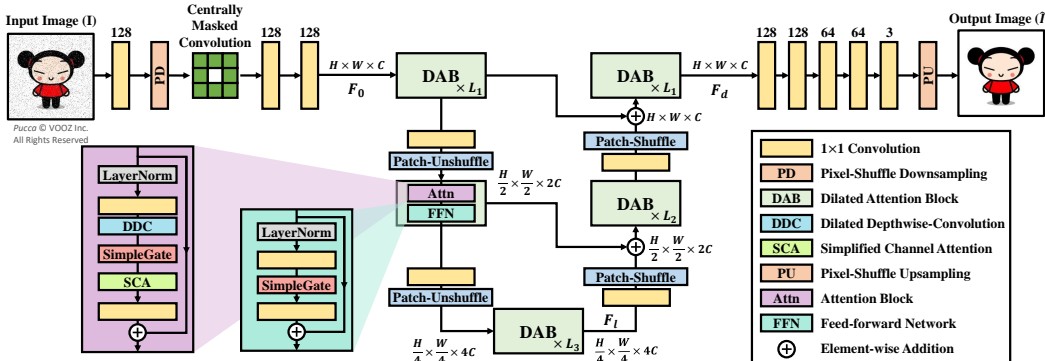

Figure 3: **Overview of PUCA.** Our method utilizes an encoder-decoder-based U-Net architecture. During encoding, we extract global context through patch-unshuffling, and during decoding, we combine local and global context through skip connections.

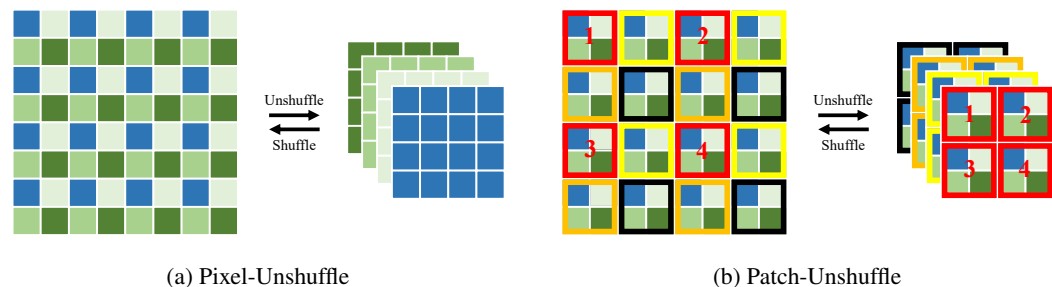

(a) Pixel-Unshuffle                         (b) Patch-Unshuffle

Figure 4: **Difference between pixel-unshuffle/shuffle and patch-unshuffle/shuffle.** The pixels except for blue are dependent on the central pixel of input, and only blue pixels are independent. We assume 2-dilated convolution is applied.

**PUCA Overview**   First, $1 \times 1$ convolution and pixel-shuffle downsampling are applied to break noise correlation in the noisy image $I \in \mathbb{R}^{H \times W \times 3}$. Subsequently, centrally masked convolution and multiple $1 \times 1$ convolutions derive low-level feature embeddings $F_0 \in \mathbb{R}^{H \times W \times C}$. $H \times W$ represents the spatial dimensions, and $C$ denotes the number of channels. The initial features $F_0$ are then processed through a 3-level encoder-decoder structure with multiple DABs at each level, resulting in deep features $F_d \in \mathbb{R}^{H \times W \times C}$. The encoder begins with the high-resolution input and incrementally shrinks its spatial size while augmenting channel capacity. Conversely, the decoder starts with low-resolution latent features $F_l \in \mathbb{R}^{\frac{H}{4} \times \frac{W}{4} \times 4 \cdot C}$ and progressively restores high-resolution representations. Patch-unshuffle and patch-shuffle operations (Figure 4b) are used for feature downsampling and upsampling, respectively. To retain detailed structures and textures in the restored images, skip connections merge features from the encoder and decoder. Finally, a sequential application of $1 \times 1$ convolution layers generates the output image $\hat{I} \in \mathbb{R}^{H \times W \times 3}$, which represents the enhanced and denoised version of the original noisy image.

### 3.1   Patch Unshuffle/Shuffle

We propose a patch unshuffle/shuffle that dramatically increases receptive fields, as shown in Figure 5. Downsampling helps to capture long-range dependencies and multi-scale context. By incorporating multi-scale context through downsampling, denoisers can better understand the relationships and dependencies between different image parts. Among the downsampling methods, pixel-unshuffle/shuffle [28] that preserves original pixels is known to be effective in image restoration. However, pixel-unshuffle with BSN [36] disrupts $\mathcal{J}$-invariance.

Let's consider BSN denoted as $f$, composed of fully convolutional layers. $f$ comprises $d$-dilated convolutions $f^{(l)}$ for all $l \in (1, L)$, with a kernel size of $3 \times 3$. We can express the function $f$ as a sequential application of $f^{(l)}$, resulting in $f(x) = f^{(L)}(f^{(L-1)}(...f^{(1)}(f^{(0)}(x))))$. Here,

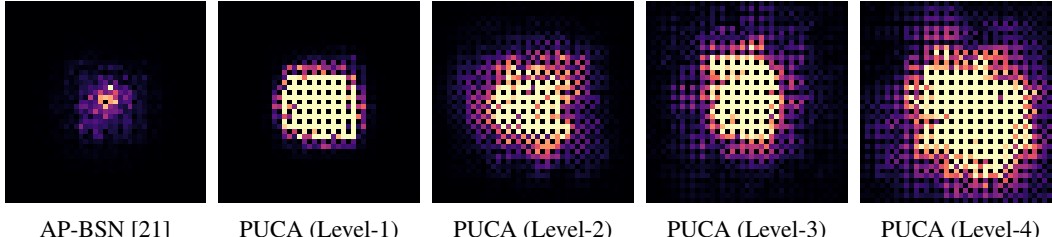

| AP-BSN [21] | PUCA (Level-1) | PUCA (Level-2) | PUCA (Level-3) | PUCA (Level-4) |

Figure 5: **Visualization of the receptive field for AP-BSN [21] and PUCA based on depth variations.** We calculate the influence of input pixels on the output's central pixel through gradients. Brighter colors indicate a stronger influence PUCA exhibits significantly wider receptive fields than AP-BSN. We observe that the receptive fields become wider as the depth increases.

$f^{(0)}$ represents a convolutional layer using a $(2d-1) \times (2d-1)$ centrally masked filter, and $x$ denotes the input noisy image. The output features for each convolutional layer $l$ are represented as $y^{(l)} = f^{(l)}(f^{(l-1)}(...f^{(1)}(f^{(0)}(x)))))$.

Let $x_{i,j}$ a pixel of a noisy image $x$, where $i, j$ are the coordinates satisfying $i \mod q = 0$ and $j \mod q = 0$. $q$ denotes the scale factor of pixel-unshuffle. The application of dilated convolution exposes $x_{i,j}$'s features to its neighboring pixels. Pixel-unshuffle (Figure 4a) aligns pixels $y_{m,n}^{(l)}$, where $i \leq m < i + q$ and $j \leq n < j + q$ to the channel axis of the same spatial position as $y_{i,j}^{(l)}$, and convolution operations following this would expose $x_{i,j}$'s features to $y_{i,j}^{(l)}$, breaking the $\mathcal{J}$-invariance.

We propose patch-unshuffle to solve the problem of pixel-unshuffle mentioned above. Patch-unshuffle transforms a tensor of size $H \times W \times C$ into a reshaped tensor of size $\frac{H}{p} \times \frac{W}{p} \times C \cdot p^2$, as illustrated in Figure 4b. This process can be mathematically defined as follows:

$$\text{Patch-Unshuffle}(y_{i,j,k}^{(l)}, p) = (p\lfloor \frac{i}{p^2}\rfloor + i \mod p, p\lfloor \frac{j}{p^2}\rfloor + j \mod p, C(p\lfloor \frac{i \mod p^2}{p}\rfloor + \lfloor \frac{j \mod p^2}{p}\rfloor) + k)$$
$$(2)$$

Incorporating patch-unshuffle in the function $f(x)$ preserves $\mathcal{J}$-invariance under specific conditions.

**Proposition 2.** Patch-unshuffle maintains $\mathcal{J}$-invariance if $p$, patch size, is a multiple of the dilation $d$ used in dilated convolution

**Proof.** Assuming a simplified one-dimensional case for clarity without loss of generality, the satisfaction of $\mathcal{J}$-invariance requires patch-unshuffle to meet the condition where pixels with the same position as $x_J$ in each channel remain unaffected by $x_J$, $J \in \{1, ..., m\}$.

After central masked convolution in the first layer, neighboring pixels are affected by $x_J$, with the receptive field $\text{RF}(y^{(0)}, x_J) = \{J - (d-1), ..., J-1, J+1, J+(d-1)\}$. $\text{RF}(y^{(l)}, x_J)$ indicates the receptive field of $x_J$ in $y^{(l)}$. Sequentially, dilated convolution spreads the information of $x_J$.

$$\text{RF}(y^{(l)}, x_J) = \bigcup_{j \in \{-d, 0, d\}} \{i + j | i \in \text{RF}(y^{(l-1)}, x_J)\} \quad (3)$$

There exist pixels that do not depend on $x_J$ with a period of $d$. To downsample $y^{(l)}$, we use patch-unshuffle. To facilitate comprehension, our emphasis lies on spatial location while disregarding the channel axis.

$$\text{Patch-Unshuffle}(y_i^{(l)}, p) = p\lfloor \frac{i}{p^2}\rfloor + i \mod p \quad (4)$$

The pixels $s$ that have identical spatial position with $J$ satisfy

$$p\lfloor \frac{s}{p^2}\rfloor + s \mod p = p\lfloor \frac{J}{p^2}\rfloor + J \mod p \quad (5)$$

$$|p(\lfloor \frac{s}{p^2}\rfloor - \lfloor \frac{J}{p^2}\rfloor)| = |(J-s) \mod p| \quad (6)$$

In Equation 6, the left-hand side is a multiple of $p$, and the right-hand side is the remainder when divided by $p$, making it smaller than $p$. Thus, for the equation to be valid, both sides must be 0 so that the positional difference between $s$ and $J$ is a multiple of $p$ from $J$. By considering the periodicity $d$ of pixels in $y^{(l)}$ that do not depend on $x_J$, resulting from the combination of centrally masked convolution and dilated convolution, we can infer that when the patch size $p$ is a multiple of the dilation factor $d$, patch-unshuffle guarantees the independence on $x_J$ along the channel axis. $\square$

By employing patch-unshuffle with a patch size $p$ that is a multiple of the dilation factor $d$, $\mathcal{J}$-invariant networks ensure the preservation of $\mathcal{J}$-invariance. This successful integration of patch-unshuffle in BSN allows for effective downsampling operations. Moreover, patch-shuffle in Figure 4b, which acts as the reverse operation of patch-unshuffle, facilitates the upsampling of downsampled feature maps to restore them to their original image sizes.

## 3.2 Dilated Attention Block

We also adopt channel attention [7] in addition to U-Net to fully exploit the global information in the image. The feature interaction that can be obtained from each is different. While channel attention based on global pooling extracts the most prominent features among the global information, U-Net is more suited for leveraging feature relations by expanding the receptive fields. However, directly applying channel attention to BSN is not straightforward due to the blind spot requirement. We aim to design a channel attention block that satisfies the blind spot requirement when combined with patch-unshuffle and patch-shuffle operations. To fulfill this requirement, we incorporate a $d$-dilated $3 \times 3$ depth-wise convolution (DDC) before gating and attention, taking inspiration from D-BSN [36]. The resulting dilated attention block (DAB), consists of LayerNorm [3], $1 \times 1$ convolution, skip connection [14], SimpleGate and Simplified Channel Attention (SCA) [7], along with DDC (as shown in Figure 3). The utilization of SimpleGate and SCA enhances the integration of local and global information by suppressing less informative features [42]. For the input feature map $\mathbf{X}$

$$\text{SimpleGate}(\mathbf{X}_1, \mathbf{X}_2) = \mathbf{X}_1 \odot \mathbf{X}_2, \tag{7}$$

where $\odot$ is element-wise multiplication, $\mathbf{X}_1$ and $\mathbf{X}_2$ are two parts of $\mathbf{X}$ along the channel axis of the same size.

$$\text{SCA}(\mathbf{X}) = \mathbf{X} * \mathcal{F}(P_\downarrow(\mathbf{X})), \tag{8}$$

where $P_\downarrow$, $\mathcal{F}$, and * denotes global average pooling, a fully-connected layer, and channel-wise product operation, respectively.

# 4 Experiments

## 4.1 Datasets and Implementation Details

**Smartphone Image Denoising Dataset (SIDD) [1]** is a collection of real-world images for denoising captured by five different smartphone cameras. Specifically, the SIDD-Medium dataset consists of 320 pairs of noisy and clean images for training purposes. In addition, the SIDD validation set and benchmark set are used for validation and evaluation, respectively. Both sets consist of 1,280 noisy patches with a size of $256 \times 256$, and corresponding clean images are provided only for the validation set. We train our model for SIDD validation and evaluation on the SIDD-medium dataset.

**Darmstadt Noise Dataset (DND) [26]** is a dataset used for benchmarking image denoising algorithms. It contains 50 real-world noisy images without any ground truth provided, and the only way to obtain results is through an online submission system. A fully self-supervised learning approach can be used to train the denoising model directly on the test set without using any external data. We train our model for DND evaluation on the DND benchmark.

**Implementation Details** We trained the model using an NVIDIA TESLA P100 GPU and implemented it with Pytorch 2.0.0. The model was trained with L1 loss between the input noisy image and the output, using Adam optimizer with an initial learning rate of 1e-4. We trained the model for 20 epochs until it fully converged. More detailed information can be found in our supplementary material. We use peak signal-to-noise ratio (PSNR) and structural similarity (SSIM) [34] as evaluation metrics for denoising. The SIDD and DND benchmark results are obtained from online submission scores, and for the SIDD validation data, we evaluate the performance using the corresponding functions in the `skimage.metrics` library.

Table 1: **Quantitative comparison on SIDD [1] and DND [26] benchmarks** While certain supervised methods achieve better evaluation results by utilizing noisy-clean image pairs from SIDD, our approach solely relies on noisy sRGB images. We report the official results from the SIDD and DND benchmark websites. Results marked with **R** and **A** are reported from R2R [25] and AP-BSN [21], respectively, while * indicates results evaluated by us.

| | Method | SIDD | | DND | |
| --- | --- | --- | --- | --- | --- |
| | | PSNR(dB) | SSIM | PSNR(dB) | SSIM |
| Non-learning based | BM3D [9] | 25.65 | 0.685 | 34.51 | 0.851 |
| | WNNM [12] | 25.78 | 0.809 | 34.67 | 0.865 |
| Supervised (Synthetic pairs) | DnCNN [45] | 23.66 | 0.583 | 32.43 | 0.790 |
| | CBDNet [13] | 33.28 | 0.868 | 38.05 | 0.942 |
| | Zhou et al. [47] | 34.00$^A$ | 0.898$^A$ | 38.40 | 0.945 |
| Supervised (Real pairs) | DnCNN [45] | 35.50* | 0.827* | 35.43* | 0.906* |
| | AINDNet (R) [16] | 38.84 | 0.951 | 39.34 | 0.952 |
| | VDN [40] | 39.26 | 0.955 | 39.38 | 0.952 |
| | DANet [41] | 39.43 | 0.956 | 39.58 | 0.955 |
| | NAFNet [7] | 40.30 | 0.961 | - | - |
| Unsupervised (Unpaired) | GCBD [6] | - | - | 35.58 | 0.922 |
| | C2N [15] + DIDN [39] | 35.35 | 0.937 | 37.28 | 0.924 |
| | D-BSN [36] + MWCNN [23] | - | - | 37.93 | 0.937 |
| Self-supervised | Noise2Void [18] | 27.68$^R$ | 0.668$^R$ | - | - |
| | Noise2Self [4] | 29.56$^R$ | 0.808$^R$ | - | - |
| | NAC [38] | - | - | 36.20 | 0.925 |
| | R2R [25] | 34.78 | 0.898 | - | - |
| | AP-BSN [21] | 35.97 | 0.925 | 38.09 | 0.937 |
| | **PUCA (Ours)** | **37.54** | **0.936** | **38.83** | **0.942** |

## 4.2 Real World Denoising

We trained with real-world images in a self-supervised way and validate our method using the popular SIDD and DND benchmark datasets. Table 1 shows a quantitative comparison of various methods on the SIDD and DND benchmarks with our method performing the best for self-supervised and unsupervised denoising. Unsupervised methods model noise using unpaired clean-noisy pairs, resulting in domain gap between training and testing. Self-supervised methods such as NAC [38] and R2R [25] are built upon less pragmatic assumptions, such as weak noise levels and known ISP function, resulting in poor performance in real-world scenarios. Our method is applicable without any restrictions as it uses sRGB noisy images for training without additional information. Figures 6 and 7 show the qualitative results for the SIDD and DND benchmarks. As shown in Figure 6, PUCA takes advantage of the global context to distinguish between meaningful features and noise showing improved denoising quality compared to the baseline methods. On the other hand, baseline methods such as AP-BSN [21] fail in exploiting global context due to the limited receptive field with notable noise remaining. PUCA, which utilizes the U-Net structure, leverages multi-scale representation capturing features at different scales. This enables the extraction of rich contextual information and facilitates the comprehension of overall object structures and their surrounding environments. As demonstrated in Figure 7, PUCA exhibits remarkable enhancements both globally and locally, specifically for complex images containing multiple objects, beyond mere flat images.

## 4.3 Ablation Study

As the number of levels in the U-Net increases, the receptive field becomes wider. Table 2 shows quantitative results according to the number of levels of the U-Net. We built several variations of our U-Net with different numbers of levels. Table 2 shows that PSNR and SSIM increase as the model gets deeper. However, in the 4-level architecture, PSNR and SSIM drop compared to level-3. This phenomenon is presumed to occur because pixel-

Table 2: Ablation study on PUCA level with SIDD validation [1].

| Level | PSNR | SSIM |
| --- | --- | --- |
| 1 | 36.768 | 0.875 |
| 2 | 37.231 | 0.878 |
| **3** | **37.492** | **0.880** |
| 4 | 36.910 | 0.878 |

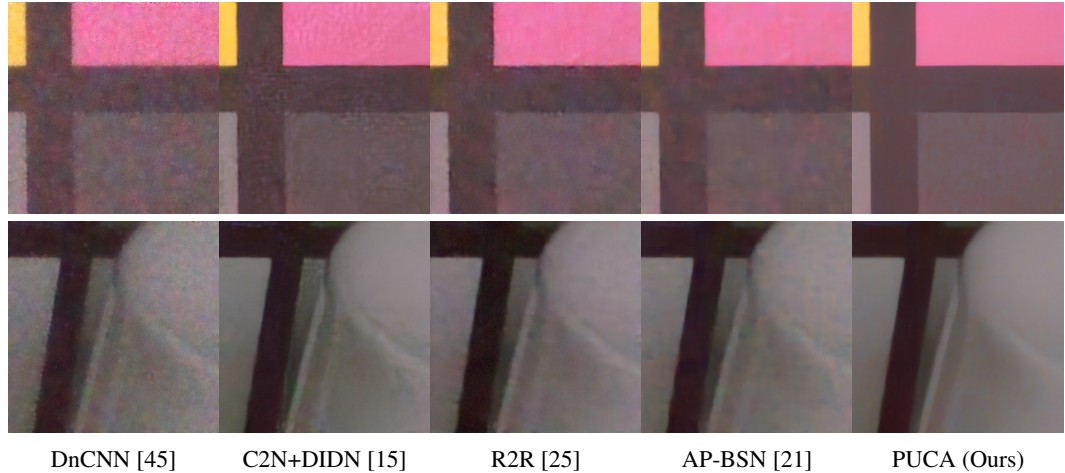

| DnCNN [45] | C2N+DIDN [15] | R2R [25] | AP-BSN [21] | PUCA (Ours) |

Figure 6: **Qualitative comparison on SIDD benchmark [1].** Access to the ground truth image and evaluation values for a denoised image is unavailable.

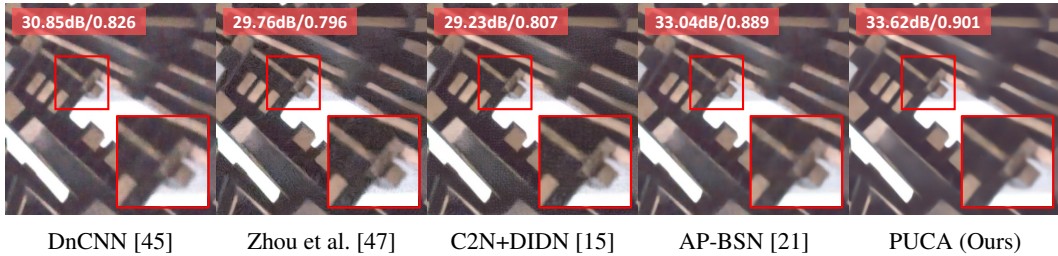

| DnCNN [45] | Zhou et al. [47] | C2N+DIDN [15] | AP-BSN [21] | PUCA (Ours) |

Figure 7: **Qualitative comparison on DND benchmark [26].**

shuffle downsampling and patch-unshuffle drastically reduce the latent features to an extremely small resolution leading to the degradation of global semantics.

Table 3 confirms the effect of DAB and patch-unshuffle. As explained in Section 3.1, the generated output of pixel unshuffle as downsampling in a 3-layer U-Net structure is equal to the original noisy input (Table 3 (a)) because the model learns identity mapping. We run experiments with each component to test the role of DAB and patch-unshuffle. For Table 3 (b), we use a 1-level network without downsampling. For Table 3 (c), we utilize the D-BSN block instead of DAB. In accordance with our intuition, Table 3 (d), which utilizes both DAB and patch-unshuffle, exhibits the best performance compared to the ablated results.

Table 3: Ablation study on PUCA components with SIDD validation [1]

|     | DAB | Unshuffle | PSNR | SSIM |
| --- | --- | --- | --- | --- |
| (a) | -   | Pixel | 23.662 | 0.328 |
| (b) | ✓   | -     | 36.768 | 0.875 |
| (c) | -   | Patch | 37.386 | **0.880** |
| (d) | ✓   | **Patch** | **37.492** | **0.880** |

## 5 Related Work

### 5.1 Image Denoising

**Supervised Image Denoising with Synthetic Pairs**  Since the success of CNNs [17, 29, 14], deep learning-based methods with CNNs have dominated the field of computer vision, including image denoising. DnCNN [45] first adopted the CNN architecture to image denoising, training on synthesized image pairs. Following this pioneering work, many studies have aimed to build a better architecture for image denoising [23, 24, 46, 30]. These learning-based methods rely on clean-noisy image pairs for training, where noisy images are synthesized with additive white Gaussian noise (AWGN). Recently, Guo et al. [13] has shown that this assumption of AWGN does not hold in

practice, and these methods show poor generalization performance due to the domain gap between real and synthetic noise.

**Unsupervised Image Denoising** Various approaches have been introduced to deal with the mentioned domain gap between training and testing in image denoising. One straightforward approach uses the clean-noisy pairs captured from real-world [1, 5]. However, this approach is impractical since data collection is extremely labor-intensive and requires a sophisticated process.

Another line of work is synthesizing realistic clean-noisy image pairs. Guo et al. [13] considers the in-camera Image Signal Processing (ISP) pipeline for realistic noise synthesis. GCBD [6] adopt adversarial training [11] to model the noise distribution and build a paired dataset for training. C2N [15] takes the properties of real-world noise into account in noise simulation.

Due to the notorious difficulty of acquiring clean-noisy pairs, self-supervised methods without clean images have been proposed. Noise2Noise [22] showed that denoising models could be trained without clean images. Noise2Void [18] and Noise2Self [4] use a masking strategy for self-supervision. The concept of blind-Spot Networks (BSNs) [18] was further improved by Laine et al. [20] and Wu et al. [36]. On the other hand, there have been some works on loss functions [37, 32] instead of explicitly masking to prevent learning identity mapping. These methods are all based on a strong assumption that noise is pixel-wise independent. Yet, they eventually learn identity mapping when applied to real-world scenarios where noise is spatially correlated. Zhou et al. [47] suggests pixel-shuffle downsampling (PD) to break the spatial correlation of real noise so that models trained with AWGN could adapt well. AP-BSN [21] enhances this with asymmetric strides during training and testing. LG-BPN [35], a contemporary work to ours, proposes a denser sampling kernel to recover local texture better and a global branch with larger receptive fields.

## 5.2 Long Range Dependency

Large receptive fields are important to extract meaningful features in consideration of context. Stacking convolutional layers can linearly expand the receptive fields of a network. Yet, downsampling operation enables exponential growth of receptive fields and thus is a more favored option in increasing receptive fields. U-Net [27] is a representative structure used in low-level vision, which employs downsampling and upsampling and it enables long-range interaction of pixels. Also, the effectiveness of U-Net's coarse-and-fine representation has been demonstrated by previous works [2, 8, 19, 33, 41, 43, 44]. Self-attention [31], another form of handling long-range dependencies, was first introduced in natural language processing (NLP) and it has been known to be also effective in computer vision [10]. Although self-attention has shown promising results recently, it has a critical shortcoming of complexity quadratic to the input resolution. Restormer [42] proposes to operate self-attention on the channel dimension rather than the spatial dimension in order to capture global information while saving computational cost. NAFNet [7] shows that simple gating and channel attention is sufficient to incorporate global context with better efficiency and performance. Following these lines of work, we aimed to apply these findings to self-supervised image denoising incorporating global context and coarse-and-fine features of large receptive fields.

## 6  Conclusion

We present PUCA, a novel $\mathcal{J}$-invariant U-Net tailored for self-supervised image denoising. By incorporating patch-unshuffle/shuffle and dilated attention block (DAB), we successfully alleviate the constrained architecture design imposed by $\mathcal{J}$-invariance. First, we propose a novel downsampling technique called patch-unshuffle, which plays a vital role in leveraging multi-scale representation and significantly expanding the receptive fields. Second, we propose DAB, which integrates global context and suppresses less informative features. The experimental results demonstrate the superiority of PUCA over existing self-supervised image denoising methods. Despite achieving state-of-the-art performance in self-supervised image denoising, PUCA still has certain limitations. One such limitation is the degradation of global semantics when using an excessive number of levels. One potential approach to address this issue is to explore methods that can break input noise correlation, thus preserving the resolution of the input. In terms of societal impacts, like other denoising methods, PUCA could be misused to invade privacy or make incorrect diagnoses.

# 7 Acknowledgements

This work was supported by the National Research Foundation of Korea(NRF) grant funded by the Korea government(MSIT) (2022R1A3B1077720 and 2022R1A5A708390811), Institute of Information & communications Technology Planning & Evaluation (IITP) grant funded by the Korea government(MSIT) [2021-0-01343: Artificial Intelligence Graduate School Program (Seoul National University) and 2021-0-02068: Artificial Intelligence Innovation Hub] and the BK21 FOUR program of the Education and Research Program for Future ICT Pioneers, Seoul National University in 2023.

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
