# Supplementary Material
# PUCA: Patch-Unshuffle and Channel Attention for Enhanced Self-Supervised Image Denoising

**Hyemi Jang[1], Junsung Park[1], Dahuin Jung[1], Jaihyun Lew[2], Ho Bae[3,∗], Sungroh Yoon[1,2,∗]**

[1]Department of Electrical and Computer Engineering, Seoul National University
[2]Interdisciplinary Program in Artificial Intelligence, Seoul National University
[3]Department of Cyber Security, Ewha Womans University

wkdal9512@snu.ac.kr    jerryray@snu.ac.kr    annajung0625@snu.ac.kr
fudojhl@snu.ac.kr       hobae@ewha.ac.kr        sryoon@snu.ac.kr

## 1  Implementation details

### 1.1  Training

We use the SIDD-Medium and DND datasets for training. For SIDD validation and SIDD benchmark evaluation, we utilize a model trained on the SIDD-Medium dataset, while for DND benchmark evaluation, we use a model trained on the DND benchmark dataset. To train our PUCA, we employ randomly cropped patches of size $256 \times 256$ from the SIDD-Medium and DND datasets. Each epoch involves 24,542 patches from the SIDD-Medium dataset and 24,784 patches from the DND dataset. To enhance the diversity of our training samples, each sample is augmented with a random $90°$ rotation and horizontal/vertical flips. Our minibatch consists of two augmented samples. The learning rate of the Adam optimizer is reduced by a factor of 10 every 8 epochs. The SIDD training took approximately 44.38 hours, while the DND training took approximately 44.78 hours.

### 1.2  Network

PUCA has a 3-level U-Net structure, with $L_1$=3, $L_2$=4, and $L_3$=4 DAB blocks at each level. From the observations of AP-BSN [3], it was confirmed that comparable denoising results were obtained when the stride factor of pixel-shuffle downsampling had a value between 4 and 6 during training. To destroy noise correlations while minimizing degradation of the global semantic structure due to the combined effects of pixel-shuffle downsampling and patch-unshuffle, we select a stride factor of 4. The stride factor of test time is 2. For the post-processing strategy, we adopt the random-replacing refinement ($R^3$) method introduced in Lee et al. [3] and set T = 8 and p = 0.16. We implement PUCA based on the AP-BSN code (https://github.com/wooseoklee4/AP-BSN.git).

### 1.3  Receptive Field Visualization

In this section, we explain the implementation of the receptive fields visualized in Figure 5 of the main text. Similar to a saliency map, we calculate the influence that the input pixel has on the central pixel of the output. We use the sum of the channels of the output's central pixel as the score and compute the gradient with respect to the input pixel of the score. The gradient indicates how much the score changes when the pixel undergoes a small modification. We take the absolute value of the gradient and then calculate the influence by finding the maximum value along the channel axis. To accentuate the differences between the receptive fields of AP-BSN and each level, we scale the calculated values by a factor of 50 and clip them to a range between 0 and 3.

---

∗Corresponding Authors

37th Conference on Neural Information Processing Systems (NeurIPS 2023).

## 2 Additional Qualitative Results

We visualize the denoising results for additional samples. The baseline models used for comparison are as follows, and the samples from these baseline models are generated from the checkpoints they provide.

Zhou et al. [6] trains a denoising model from additive white Gaussian noise and applies the model to real noisy images where the noise correlation is broken by pixel-shuffle downsampling with a stride of 2. C2N trains a generator to create pseudo-noisy images and then trains a denoising model like DIDN using noisy-clean pairs (C2N+DIDN) [2]. R2R [4] trains the denoising models using noisy pairs created from unorganized noisy images. AP-BSN [3] adopts asynchronous pixel-shuffle downsampling to train the denoising model to minimize loss of detail while breaking the noise correlation.

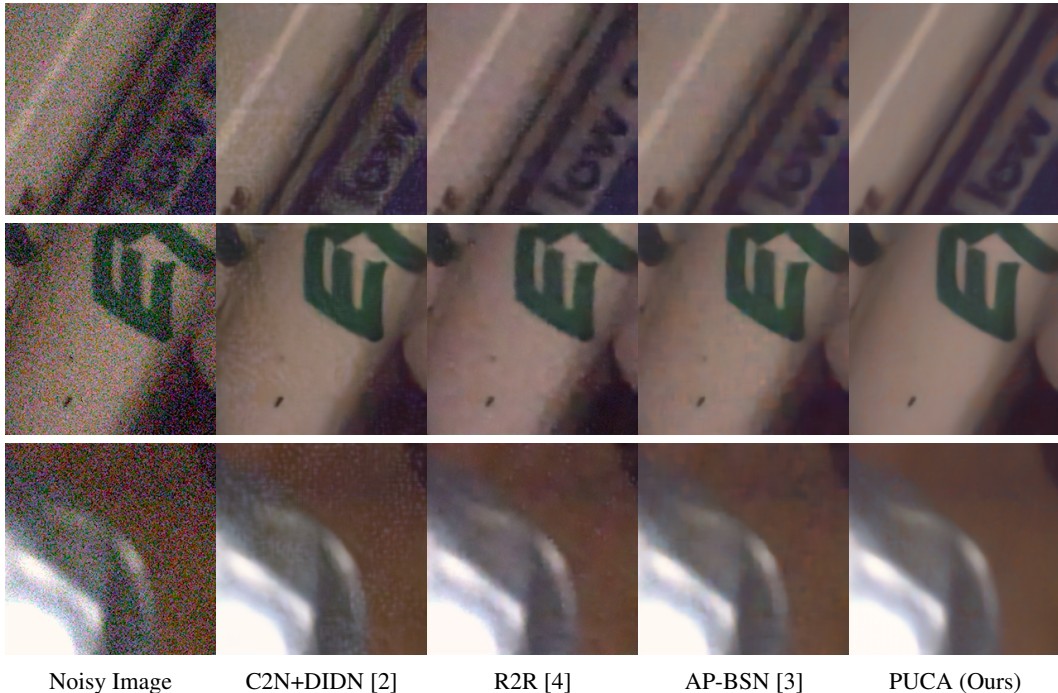

Noisy Image      C2N+DIDN [2]      R2R [4]      AP-BSN [3]      PUCA (Ours)

Figure 1: **Additional qualitative comparisons on SIDD benchmark [1]**.

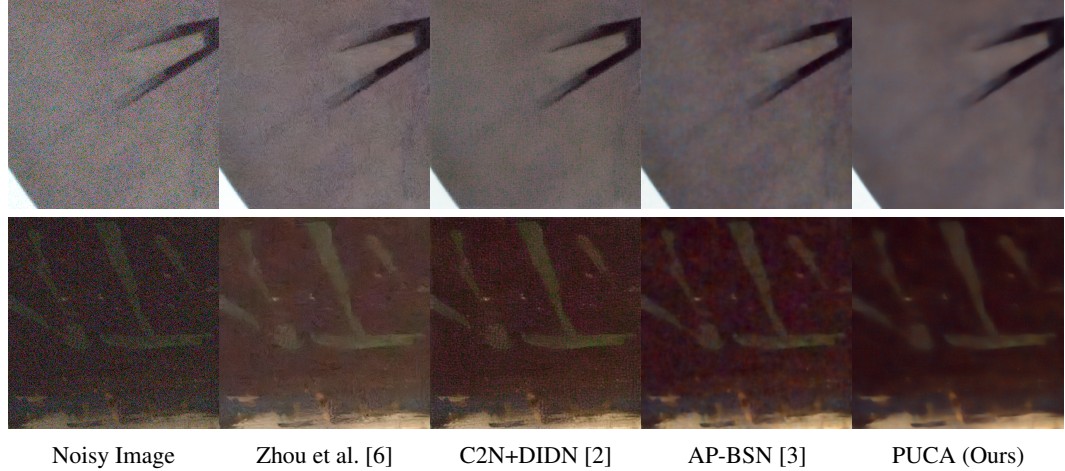

Noisy Image      Zhou et al. [6]      C2N+DIDN [2]      AP-BSN [3]      PUCA (Ours)

Figure 2: **Additional qualitative comparisons on DND benchmark [5].**

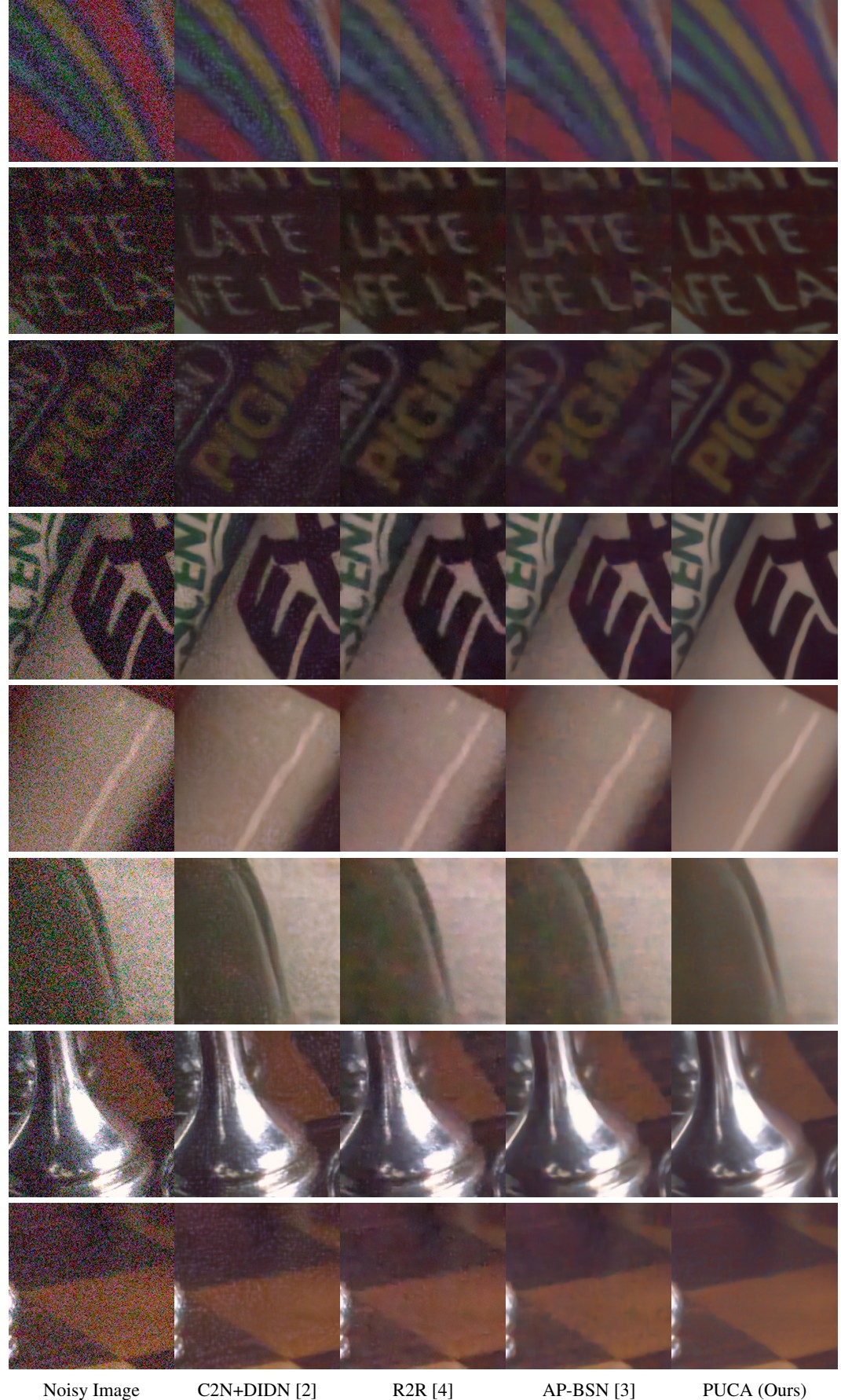

Noisy Image        C2N+DIDN [2]        R2R [4]        AP-BSN [3]        PUCA (Ours)

Figure 3: **Additional qualitative comparisons on SIDD validation [1].**