# OpenReview forum: "PUCA: Patch-Unshuffle and Channel Attention for Enhanced Self-Supervised Image Denoising"
_NeurIPS.cc/2023/Conference — NeurIPS 2023 poster_

### Official Review · Reviewer_3t6u · 2023-06-14

**Soundness:** 3 good
**Presentation:** 3 good
**Contribution:** 2 fair
**Rating:** 7
**Confidence:** 5

**Summary:**

This paper propose a novel network architecture for self-supervised image denoising. The network is build with foundation blocks of NAFNet, which combines mask and dilated convolutions to implement the blind-spot network. Patch-unshuffle is introduced as downsample/upsample operations with enables multi-scale design of the network while maintaining the blind-spot mechanism. Experimental results show the effectiveness of proposed method on real-world denoising datasets.

**Strengths:**

1. The idea of searching for flexible network architecture is reasonable for improving the self-supervised denoising performance.
2. Patch-unshuffle is well designed to enable downsample/upsample operations for implementing multi-scale network architecture.
3. Experimental results show the effectiveness of proposed method in self-supervised image denoising on real-world images.
4. The paper is well written and easy to follow.

**Weaknesses:**

1. The overall novelty is limited. The training process is the same as AP-BSN, and the network architechture is assembled with existing DBSN and NAFNet. The only contribution is the idea of improving the network flexibility and to implement a multi-scale network with patch-unshuffle, which is not technically sound.
2. The ablation study is not complete. The central idea of this paper is to improve DBSN used in AP-BSN with multi-scale architecture, however, there are existing multi-scale BSN architechture such as [1]. What's the performance of porposed network compare with [1]?
3. The competing methods is not complete. This paper shows good performance on real-world image denoising, but the latest self-supervised denoising methods [2][3] are ignored.

[1] High-quality self-supervised deep image denoising. NIPS, 2019.

[2] LG-BPN: Local and Global Blind-Patch Network for Self-Supervised Real-World Denoising. CVPR, 2023.

[3] Spatially Adaptive Self-Supervised Learning for Real-World Image Denoising. CVPR, 2023

**Questions:**

My major concern is weakness 2 and 3, please complete the necessary experiments.

**Limitations:**

The authors hav adequately addressed the limitations and potential negative societal impact.

---

> ### Author Rebuttal · Authors · 2023-08-09
>
> **W1**
> The overall novelty is limited. The training process is the same as AP-BSN, and the network architechture is assembled with existing DBSN and NAFNet. The only contribution is the idea of improving the network flexibility and to implement a multi-scale network with patch-unshuffle, which is not technically sound.
>
> **WA1**
> As you have mentioned in the strengths section, we devised multi-scale network architectures through patch-unshuffle/shuffle to enhance self-supervised denoising performance. This outcome emerges from the expansion of network design, which was previously constrained due to the requirements of BSN. Additionally, by incorporating the J-invariant block with channel attention, known as DAB, we achieved an ensemble effect from the subsamples generated by patch-unshuffle. This successful integration allowed us to effectively elevate denoising performance.
>
> ---
>
> **W2**
> The ablation study is not complete. The central idea of this paper is to improve DBSN used in AP-BSN with multi-scale architecture, however, there are existing multi-scale BSN architechture such as [1]. What's the performance of porposed network compare with [1]?
>
> **WA2**
> Based on our understanding of [1] you referred to, it appears that all convolutional layers employ a shifted upwards kernel with zero values below the center row and incorporate offsets. The same approach is used for downsampling/upsampling, utilizing offset methods along with either average pooling or nearest-neighbor upsampling. It seems plausible that for ablation studies: 1) DAB could be replaced with the proposed convolution and 2) patch-unshuffle/shuffle could be substituted with offset-incorporated avg. pooling/nearest-neighbor upsampling.
> In [1], the maintenance of j-invariant is achieved through a combination of downward rows with zero-value convolutions, offsets, and avg. pooling/nearest-neighbor upsampling. Our method is optimized for centrally masked convolutions and dilated convolutions, which is why using the alternative approach that we suggested could potentially break the j-invariant. Consequently, both methods might yield results similar to pixel-unshuffle/shuffle.
>
> ---
>
> **W3**
>  The competing methods is not complete. This paper shows good performance on real-world image denoising, but the latest self-supervised denoising methods [2][3] are ignored.
>
> **WA3**
> At the time of our proposed method's submission, neither [2] nor [3] had been published. However, in response to your request, we conducted a comparison.
> LG-BPN [2] augmented receptive fields through expanded convolutions and a local-global branch. SASL [3] employed a Blind-Neightborgood Network and a Locally Aware Network to separate denoising into textured and flat regions.
> PUCA effectively expands the receptive field by downsampling/upsampling feature maps using patch-unshuffle/shuffle. Furthermore, by extracting global context from a multi-scale representation and integrating fine details through skip connections, PUCA benefits denoising without the need for distinct regions or local-global separation, unlike other methods.
> The performance on the SIDD and DND benchmarks is as follows, with results from [2] and [3] extracted from the original texts:
>
> |             | SIDD benchmark (PSNR/SSIM) | DND benchmark (PSNR/SSIM) |
> |-------------|----------------------------|---------------------------|
> | PUCA (Ours) | 37.54/0.936                | 38.83/0.942               |
> | LG-BPN [2]  | 37.28/0.936                | 38.43/0.942               |
> | SASL [3]    | 37.41/0.934                | 38.18/0.938               |
>
> Thank you for your suggestion. we will incorporate it into the final version.

---

> > ### Comment · Reviewer_3t6u · 2023-08-17
> >
> > The rebuttal has addressed most of my concerns, and I'll improve rating from 6 to 7.

---

> > > ### Author Response · Authors · 2023-08-17
> > > **Thank you for updating the score!**
> > >
> > > We greatly appreciate the insightful feedback you've provided. Your raised points have prompted us to reevaluate the approaches we had overlooked. Your feedback has been instrumental in improving the manuscript, and we sincerely thank you for raising the score to 7. Your perspective has notably elevated the quality of the paper, and we're open to any additional thoughts or suggestions you want to share.

---

### Official Review · Reviewer_jJKN · 2023-07-02

**Soundness:** 3 good
**Presentation:** 3 good
**Contribution:** 3 good
**Rating:** 7
**Confidence:** 3

**Summary:**

This work addresses the problem of developing a self-supervised learning-based denoiser. To this end, this work proposes the concepts of patch shuffle/unshuffle operations to effectively downsample and upsample the features while ensuring that the J-invariance property holds with the resulting network. Using dilated attention blocks and patch shuffle/unshuffle, this work comes up with a variant of Unet architecture that can significantly improve J-invariance networks' spatial information aggregation ability. The proposed method improves the denoising performance on public benchmark datasets.

**Strengths:**

1. The paper is written well.
2. The new notions of patch shuffle/unshuffle and dilated attention blocks to ensure the holding of J-invariance properly is interesting
3. The proposed method outperforms prior arts significantly

**Weaknesses:**

1. A study is missing to elaborate more on what other network architectures the proposed ideas can find a perfect fit. There have been numerous variants of Unet proposed in the literature. Can those extensions boost the performance further remain unclear.

**Questions:**

- Please address my comments under weaknesses.

- Visual comparisons with other self supervised methods (example, Figure 6) reveals as if the proposed method lacks details while removing the noise completely whereas other methods retains more details. Is there an explanation on why so? Is it possible to control the denoising level of the proposed method to earn back the lost details?

**Limitations:**

Limitations has been addressed adequately.

---

> ### Author Rebuttal · Authors · 2023-08-09
>
> **W1** A study is missing to elaborate more on what other network architectures the proposed ideas can find a perfect fit. There have been numerous variants of U-Net proposed in the literature. Can those extensions boost the performance further remain unclear.
>
> **WA1**
> Structures employing self-attention, such as Uformer [42], are left for future work due to the potential of self-attention to compromise J-invariance. Adhering to BSN's requirements, we adapted the MIMO-UNet [8]. Our training yielded results of PSNR: 37.209 dB and SSIM: 0.875 on the SIDD validation set. Due to time constraints, extensive hyperparameter searching was not conducted. However, our modified MIMO-UNet surpassed the denoising performance of AP-BSN [21]. Thus, we anticipate that PUCA could enhance performance across various U-Net variants.
>
> ---
>
> **Q1** Visual comparisons with other self supervised methods (example, Figure 6) reveals as if the proposed method lacks details while removing the noise completely whereas other methods retains more details. Is there an explanation on why so? Is it possible to control the denoising level of the proposed method to earn back the lost details?
>
> **QA1**
> Sorry for the confusion. It might seem that we have lost some details by not visualizing the ground truth together, but it is apparent from Figure 6 that PUCA indeed displays sharper edges compared to other models. Given this observation, we believe that our method does not miss the details. It seems that the baselines might preserve detail that appears more akin to less-removed noise, rather than enhancing real image details, as per our perception. Upon reviewing Figure 1 and the supplementary materials, we can confirm that PUCA represents finer details like text with greater clarity. Adjusting the level or dilation of the model does offer the possibility to control details, but along with that, denoising also becomes weaker, making it less practical.

---

> > ### Comment · Reviewer_jJKN · 2023-08-17
> > **Post-rebuttal comments**
> >
> > The rebuttal has addressed all my concerns. I retains my original rating.

---

> > > ### Author Response · Authors · 2023-08-17
> > > **Thank you for the valuable review!**
> > >
> > > We appreciate your review and we are glad to hear that the rebuttal has effectively addressed your concerns. Thank you for maintaining your original rating. Your feedback has been valuable in enhancing the quality of the manuscript.

---

### Official Review · Reviewer_DQiP · 2023-07-04

**Soundness:** 3 good
**Presentation:** 3 good
**Contribution:** 3 good
**Rating:** 6
**Confidence:** 4

**Summary:**

This work extends the field of blind spot networks (BSN) for self-supervised image denoising. They propose patch-unshuffle/shuffle a technique for downsampling/upsampling that preserve J-invariance, which allows to build U-Net like architectures expanding the receptive field and utilizing multi-scale representations in BSN.
Second, the dilated attention block is introduced, a J-invariant channel attention mechanism incorporating global information. The method outperforms existing self-supervised methods by 1.6dB (SIDD dataset) and 0.7dB (DND dataset) in PSNR resulting also in improved perceived image quality for the depicted examples.

**Strengths:**

1. The proposed method is interesting and alleviates the constrained architecture design imposed by J-invariance opening up possibilities for new designs of BSNs.
2. The experiments on the two most common benchmarks for self-supervised image denoising are extensive and the method is compared to a range of baseline methods.
3. The authors theoretically derive the J-invariance property of their proposed patch-unshuffle method.

**Weaknesses:**

1. Section 3.2 introduces the dilated attention block (DAB) that preserve J-invariance. However, from the sentence "To fulfill this requirement we incorporate a d-dilated 3x3 depth-wise convolution (DDC) before gating and attention, taking inspiration from D-BSN" it is not clear to me how exactly J-invariance is preserved nor if or how the approach differs from the previous work.
2. From Section 2.2 it is not clear to me how pixel-shuffle downsampling (PD) works. From Figure 4 it seems as PD changes the dimension of the input image, however this is not discussed in Section 2.2 and the algorithm overview in Figure 3 shows the signal before and after PD to exhibit the same dimension.
3. I find that dropping the channel dimension in the proof in Section 3.1 is a bit confusing as now several input indices are mapped to the same output index. Maybe an illustration with an explicit example of how a set of indices is transformed might be helpful.
4. Summarizing the above points the clarity of the work could be improved.

**Questions:**

1. Choosing the by far worst performing supervised baseline as the only visual reference for supervised training in Figure 1 seems a bit misleading with respect to the comparison of supervised and self-supervised methods.
2. The caption in Table 3 (3b with DAB) does not fit the description in the text (3b no DAB).
3. Line 256 refers to other work without citing it.
4. The paper uses a numeric citation style in the text but alphabetic in the references making it impossible to match citations to entries in the references.

**Limitations:**

The authors briefly touch on one limitation of their work.

---

> ### Author Rebuttal · Authors · 2023-08-09
>
> **W1** Section 3.2 introduces the dilated attention block (DAB) that preserve J-invariance. However, from the sentence "To fulfill this requirement we incorporate a d-dilated 3x3 depth-wise convolution (DDC) before gating and attention, taking inspiration from D-BSN" it is not clear to me how exactly J-invariance is preserved nor if or how the approach differs from the previous work.
>
> **WA1**
> In Figure 3 of the main text, layers of the DAB block are depicted. The DAB block integrates components including LayerNorm and 1x1 convolution, DDC, SimpleGate, and SCA. LayerNorm and 1x1 convolution do not affect j-invariance. SimpleGate performs element-wise multiplication on two chunks divided along the channel axis, thereby maintaining j-invariance similarly. SCA goes through global average pooling and a fully connected layer to create channel-attention. Then, it multiplies the input and the channel-attention along the channel axis Consequently, even in this scenario, it has no impact on j-invariance.
> In the context of the conventional channel-attention block, using simple depth-wise convolution instead of DDC disrupts the j-invariance upheld by centrally masked/dilated convolutions, as illustrated in Figure 2. To address this, DDC (a dilated depth-wise convolution) is proposed to maintain j-invariance. While D-BSN can be used as is, channel attention was introduced to harness ensemble effects from the subsamples generated by patch-shuffle/unshuffle. As indicated in Table 3 (c) and (d), the presence of DAB leads to a PSNR increase of 0.106.
>
> Table 3: Ablation study on PUCA components with SIDD validation
> |     | DAB | Unshuffle | PSNR   | SSIM  |
> |-----|-----|-----------|--------|-------|
> | (a) | -   | Pixel     | 23.662 | 0.328 |
> | (b) | V   | -         | 36.768 | 0.875 |
> | (c) | -   | Patch     | 37.386 | 0.880 |
> | (d) | V   | Patch     | 37.492 | 0.880 |
>
> ---
>
> **W2** From Section 2.2 it is not clear to me how pixel-shuffle downsampling (PD) works. From Figure 4 it seems as PD changes the dimension of the input image, however, this is not discussed in Section 2.2 and the algorithm overview in Figure 3 shows the signal before and after PD to exhibit the same dimension.
>
> **WA2**
> PUCA adopts the pixel-downsampling approach from AP-BSN [21]. Similar to the observation in Figures 4 and 5 of AP-BSN, pixel-downsampling (PD) serves to arrange the subsamples that are listed along the channel axis of pixel-unshuffle onto a single plane. Consequently, Figure 3 of the main text exhibits the same channel dimensions before and after PD.
>
> ---
>
> **W3** I find that dropping the channel dimension in the proof in Section 3.1 is a bit confusing as now several input indices are mapped to the same output index. Maybe an illustration with an explicit example of how a set of indices is transformed might be helpful.
>
> **WA3**
> We agree with your suggestion, multiple input indices mapping to the same output index could potentially lead to confusion. We visualized the operation principle of patch-unshuffle in Figure 4 (b). Patches with the same color are assembled along the same channel axis, and patches of different colors are arranged in the same order as indicated by the red patches.
>
> ---
>
> **Q1** Choosing the by far worst performing supervised baseline as the only visual reference for supervised training in Figure 1 seems a bit misleading with respect to the comparison of supervised and self-supervised methods.
>
> **QA1**
> We agree with your suggestion and we will include additional supervised models into the final version.
>
> ---
>
> **Q2** The caption in Table 3 (3b with DAB) does not fit the description in the text (3b no DAB).
>
> **QA2** As you mentioned, the table caption in the main text has indeed been changed. We willl make the necessary corrections. Thank you for bringing this to our attention.
>
> ---
>
> **Q3** Line 256 refers to other work without citing it.
>
> **QA3** We will add the citation to the work at line 256, as you suggested.
>
> ---
>
> **Q4** The paper uses a numeric citation style in the text but is alphabetic in the references making it impossible to match citations to entries in the references.
>
> **QA4** We will match the bibliography style to the number citation style. This should help avoid any confusion.

---

> > ### Comment · Reviewer_DQiP · 2023-08-16
> >
> > Thank you for the reponse and clarifying my questions regarding D-BSN vs. DAB and PD (pixel-shufle downsampling) in Section 2.2 vs. pixel-unshuffle in Figure 4a.
> > I suggest adding some of those explanations to the paper to make it more self-contained as for now it requires detailed knowledge of the concepts in [47,21] and [36] to follow the paper.
> > Through that together with including more supervised baselines as mentioned in QA1, I believe that the presentation of the paper has been improved and I increased the presentation score from 2 to 3 and the overall score from 5 to 6.

---

> > > ### Author Response · Authors · 2023-08-17
> > > **Thank you for updating the score!**
> > >
> > > Thank you for your response. Thanks to your deep insights, we can clearly distinguish between pixel shuffle downsampling (PD) and pixel unshuffle and clearly explain the difference between D-BSN and DAB.
> > > We fully agree with the suggestion to include these explanations in the body of the paper. This approach aligns well with our goal of making the paper more self-contained by increasing clarity and understanding.
> > > We also thank you for emphasizing the need to incorporate additional supervised baselines, as highlighted in QA1. Your comments have been very helpful in refining the manuscript, and we greatly appreciate your thoughtfulness in adjusting the score to a 6.
> > > We sincerely appreciate your valuable feedback and insightful suggestions. Your perspective has greatly improved the quality of the paper, and if you have any additional insights or recommendations, please feel free to share them with us.

---

### Official Review · Reviewer_daq5 · 2023-07-06

**Soundness:** 3 good
**Presentation:** 3 good
**Contribution:** 2 fair
**Rating:** 6
**Confidence:** 4

**Summary:**

This paper presents a method for self-supervised image denoising. Specifically, a patch-unshuffle (and shuffle) operation together with a dilated attention block was proposed to achieve the goal. Experimental results on two real noise datasets (SIDD and DND) show the effectiveness of the proposed method over other self-supervised methods.

**Strengths:**

+ Real noisy image denoising is an important problem in the denoising community and the authors also alleviate the limitations from the $\mathcal{J}$-invariance to some extent.

+ The proposed method was shown to perform generally better than other self-supervised methods on the SIDD and DND real noise datasets.

+ The paper is generally well-written and easy to follow.

**Weaknesses:**

- From the description, it seems the authors were motivated by the flexibility in network structures. But it is unclear why the flexibility in network structure design is necessary, or how would it benefit the denoising community.

- The novelty of the proposed patch-unshuffle/shuffle is a bit limited, especially considering the previously proposed pixel-unshuffle/shuffle approach [47]. The key idea is very similar with the only difference as pixel vs. patch. Whereas the significance of such a change and novelty were not well clarified with convincing justifications. For example, what if replace the patch-unshuffle/shuffle blocks in the proposed method with the corresponding pixel-wise version? It is also unclear what is the contribution of the PD and PU processing at the input/output ends. What if remove these processing operations and only use the proposed patch-wise blocks?
On the other hand, from what was shown in Fig.5 in [47], it seems that they already used a similar idea of patch-shuffle.

- The proposed patch-unshuffle was motivated by the claimed issue of breaking the \mathcal{J}-invariance in pixel-unshuffle. But as also acknowledged by the authors, the proposed patch-unshuffle can still break it in some cases.

- The novelty of the proposed dilated attention block is also a bit limited. As mentioned by the authors, it was based on the channel attention [7] being applied to U-Net, with inspirations from D-BSN [36] and SCA [7]. If there is a new design involved, the contribution and novelty are unclear, and it lacks convincing (experimental) validation, e.g. what if without the new design? how the performance relates to the proposed design?

- It is unclear what dilation rate was used in the proposed DAB block. And unclear how this hyper-parameter d affects the model's performance.

- The proposed method performs worse than the C2N+DIDN for the SSIM (Table 1). We know that SSIM usually represents the visual or structural quality of the reconstructed image. But there is no clarification for this.

- In the ablation study (Table 2), the authors attribute the decreased performance when the level increased from 3 to 4, to the resolution of the latent features. To validate this, an experiment with a larger input image (thus a higher resolution for the latent features when level=4) should have been included.

- Although LG-BPN [35] may not be published (and not published at CVPR) when the proposed method was submitted, the idea of local-global branches and the larger receptive fields through dilated convolutions is very similar to the proposed method. The novelty is as a result weakened.

- The proposed method claimed the issues within existing methods about the \mathcal{J}-invariance, and motivated by this, claimed that the proposed method "successfully alleviates the constrained architecture design imposed" by this. But it is unclear how the constraints for architecture design were alleviated. This was not validated, unless other designs were shown to be effective with the proposed method.

- Missing definition to p (L136-144). If it's the same as the p in L145, please specify (the first time it appears).

- It is unclear what "PUCA" indicates, "Patch-Unshuffle Channel Attention", or "xxxUnetxxx"? Please clearly define it the first time it was introduced.

- In the caption of Fig. 3, "During encoding...through patch shuffling..." should be "patch unshuffling"?

**Questions:**

It would be helpful if the authors could address the concerns raised in the above Weaknesses section. For example, the main motivation of the proposed method and its validity; the technical novelty of the newly proposed components (i.e. the patch-unshuffle and the DAB blocks); those concerns about the experiments.

The reviewer would be happy to change the rating if the concerns could be well addressed.

**Limitations:**

The authors clearly mention the limitations of their method (in the Conclusion) and potential societal impacts.

---

> ### Author Rebuttal · Authors · 2023-08-09
>
> Table 3: Ablation study on PUCA components with SIDD validation
> |     | DAB | Unshuffle | PSNR   | SSIM  |
> |-----|-----|-----------|--------|-------|
> | (a) | -   | Pixel     | 23.662 | 0.328 |
> | (b) | V   | -         | 36.768 | 0.875 |
> | (c) | -   | Patch     | 37.386 | 0.880 |
> | (d) | V   | Patch     | 37.492 | 0.880 |
>
> **W1** BSN's evolution has shifted from input-masking to a centrally-masked kernel approach for efficiency and reduced artifacts. Due to BSN's constraints, only stacked dilated convolutions were viable. In contrast, supervised denoising, like U-Net [2,8,19,33,41,42,46], enjoys more design flexibility. U-Net extends receptive fields, extracts multi-scale global context, and combines details through skip connections, making it effective. By infusing scalability into self-supervised denoising design, we foresee new network structures emerging for enhanced performance.
>
> **W2** In self-supervised denoising, the target image matches the input noisy image. To prevent identity mapping, BSNs have been used, requiring j-invariance. However, centrally masked/dilated convolutions with pixel-shuffle/unshuffle can disrupt j-invariance and hamper denoising. [47] introduced pixel-shuffle down-sampling adaptation as shown in Figure 5 of [47]. To maintain j-invariance and successful denoising, we propose patch-unshuffle/shuffle for downsampling/upsampling. Table 3 (a) in the main text highlights that pixel-shuffle/unshuffle generates identical outputs to the input image.
>
> Real noise displays correlations distinct from synthetic noise, challenging the assumption of independent zero-mean noise in synthetic methods. Thus, synthetic-based approaches struggle with real noise generalization. PD disrupts noise correlations by arranging subsamples along the channel axis onto a plane as seen in AP-BSN’s Figures 4 and 5. PU recombines subsampled images. While j-invariance remains intact without PD and PU, real noise's characteristics allow predictions influenced by neighboring pixels, resembling identity mapping. This applies to AP-BSN and LG-BPN as well.
>
> **W3** D-BSN constructs its network through a combination of centrally masked/dilated convolution. As demonstrated in Noise2Kernel*, maintaining J-invariance necessitates a dilation of dilated convolution, $d \geq \mathrm{ceil}(K/2)$ (where $\mathrm{ceil}(a)$ represents the smallest integer greater than or equal to a, and $K$ signifies the kernel size of centrally masked convolution). Similar to adjusting the dilation of dilated convolution to satisfy BSN's requirement, adjusting the patch size of patch-unshuffle serves as a method to meet the demands of BSN.
>
> **W4** Using the channel attention [7] directly would break j-invariance. Hence, we made the modification of changing the convolution in the channel attention to dilated convolution. While D-BSN could be utilized as is, we incorporated channel attention to gain ensemble effects from the subsamples generated by patch-shuffle/unshuffle. In Table 3 (c) and (d), it can be observed that the presence of DAB results in a PSNR increase of 0.106.
>
> **W5** In the main text, we utilized a dilation of 2, and in accordance with your request, the effects of experimentation with different dilations are as follows:
>
> | Dilation | PSNR   | SSIM  |
> |----------|--------|-------|
> | 2        | 37.492 | 0.880 |
> | 3        | 37.284 | 0.884 |
> | 4        | 36.824 | 0.879 |
>
>  As the dilation size increases, the PSNR decreases. We infer that this phenomenon occurs due to the simultaneous increase in dilation and patch-unshuffle size, which presents challenges in reconstructing local information.
>
> **W6**
> Our intuition is that C2N+DIDN's stabilizing loss term helps maintain the luminance and contrast of the dataset, and the addition of C2N-generated noise to the clean image enhances structural aspects, leading to high SSIM scores. Furthermore, we infer that the SIDD dataset's larger object sizes relative to the noise positively contribute to C2N+DIDN achieving high SSIM values.
>
> **W7** As per your request, we measured the performance based on level variations in the DND dataset with a size of 512x512. The results are as follows:
>
> | level   | PSNR   | SSIM  |
> |---------|--------|-------|
> | level-3 | 38.884 | 0.942 |
> | level-4 | 39.030 | 0.944 |
>
> As anticipated in the main text, it is evident that as the resolution increases, performance improves with deeper levels (level 3 to level 4).
>
> **W8** LG-BPN augmented receptive fields through expanded convolutions and a local-global branch. PUCA effectively expands the receptive field by downsampling/upsampling feature maps using patch-unshuffle/shuffle. Furthermore, by extracting global context from a multi-scale representation and integrating fine details through skip connections, PUCA benefits denoising without the need for local-global separation.
>
> **W9** Sorry for the confusion. We intended that we introduced scalability to network design. To assess the potential for extension in U-Net variants, we modified [8], considering the requirements of BSN. As a result, we obtained a PSNR of 37.209 dB and an SSIM of 0.875 on the SIDD validation. While further optimization seems necessary, we anticipate that the components of PUCA can be applied to various U-Net variants to enhance performance.
>
> **W10** With the same $p$ as in L145, we will define it according to your opinion when it first appears.
>
> **W11** "PUCA" stands for "Patch-Unshuffle and Channel Attention," and, we will make sure to properly include it in the introduction as you suggested. Thank you for the advice.
>
> **W12** We will make the correction to "Patch Unshuffling" in Figure 3 as you mentioned.
>
> [*] Noise2kernel: Adaptive self-supervised blind denoising using a dilated convolutional kernel architecture. Sensors, 2022

---

> > ### Comment · Reviewer_daq5 · 2023-08-17
> > **Re: rebuttal**
> >
> > Thanks to the authors' rebuttal. I have read the rebuttal and comments from other reviewers. It is good to see that most of my concerns were addressed, making the paper more clear and easier to understand. The authors are suggested to include these explanations into their final version. Although the technical novelty is still a bit limited to me, given the related prior works, I would raise my rating in response to the major concerns being addressed.

---

> > > ### Author Response · Authors · 2023-08-18
> > > **Thank you for updating the score!**
> > >
> > > Thank you for your response.
> > > The questions you raised have helped us to clarify the role of dilation and levels again, and we completely agree with your suggestion to add more detailed explanations throughout the paper to make it clearer and easier to understand.
> > > Your insights have been very helpful in refining the manuscript, and we deeply appreciate your thoughtful adjustment of the score to a 6. Thank you very much for your valuable feedback and insightful suggestions. Your perspective has greatly improved the quality of the paper, and if you have any additional insights or recommendations, please feel free to share them with us.

---

### Official Review · Reviewer_nv4m · 2023-07-06

**Soundness:** 4 excellent
**Presentation:** 4 excellent
**Contribution:** 3 good
**Rating:** 7
**Confidence:** 3

**Summary:**

This paper introduces PUCA, a J-invariant U-Net for self-supervised image denoising. Specifically, the authors propose a patch-unshuffle and dilated attention block to allow the use of the multi-scale structure for enlarging the receptive field. Extensive experiments demonstrate that the proposed PUCA outperforms the existing method by a notable margin. There is also adequate analysis to illustrate the correctness and properties of the proposed method.

**Strengths:**

1. Good Quality. The submission is technically sound in my opinion and the advantages and limitations of this work are discussed carefully and honestly.
2. Good Clarity. The submission is written with sufficient clear definitions and formulas. And the organizations are well-designed.
3. Good motivation. The authors observe that the commonly used simple structure, such as multi-scale structure,  in image denoising violates the J-invariance and thus cannot be used in BSNs.
4. Solid solution. The authors propose patch unsuffle to achieve downsampling without violating j-invariance. In addition, proof is also provided to demonstrate this property of patch unshuffle theoretically.
5. Sufficient experiments. The authors conduct sufficient experiments and ablation studies to demonstrate the effectiveness of the proposed method. The results show that the proposed method outperforms all existing methods by a notable margin.

**Weaknesses:**

No obvious weaknesses.

**Questions:**

In L28, the authors emphasize the generalization issue of the supervised method and note that the unsupervised method can generalize better. However, this claim is not justified. Can the author show the cross-dataset validation of both the supervised method and the unsupervised method?

**Limitations:**

The paper contains an adequate discussion of social impacts and limitations.

---

> ### Author Rebuttal · Authors · 2023-08-09
>
> **Q1**
>  In L28, the authors emphasize the generalization issue of the supervised method and note that the unsupervised method can generalize better. However, this claim is not justified. Can the author show the cross-dataset validation of both the supervised method and the unsupervised method?
>
> **QA1**
> Sorry for the confusion. The meaning of L28 in the main text was that supervised denoising approaches require extreme cost in data collection, making it challenging to gather large-scale data, which would inevitably lead to limited generalization ability. We will revise the main text to resolve this confusion. In line with your request, we conducted cross-dataset experiments (testing the model trained on SIDD with the DND benchmark) and obtained the following reasonable results:
>
> |                              | SIDD (PSNR/SSIM) | DND (PSNR/SSIM) |
> |------------------------------|------------------|-----------------|
> | Restormer (SIDD trained)[43] | 40.02/0.960      | 40.03/0.956     |
> | PUCA (SIDD trained)          | 37.54/0.936      | 38.60/0.940     |
>
> The results of Restormer, a type of supervised image denoiser, are from the results reported in the original paper [43]. We would like to emphasize the significance of our self-supervised image denoiser demonstrating comparable performance to the supervised image denoiser.

---

> > ### Comment · Reviewer_nv4m · 2023-08-17
> > **Post-rebuttal Comments**
> >
> > Thanks for the authors' feedback. It has addressed my question. I'll maintain my original rating.

---

> > > ### Author Response · Authors · 2023-08-17
> > > **Thank you for the valuable review!**
> > >
> > > Thank you for your review, and we are pleased to hear that the concerns have been effectively addressed through the rebuttal. We appreciate your decision to maintain the original rating. Your feedback has been instrumental in improving the quality of the manuscript.

---

### Author Rebuttal · Authors · 2023-08-09

We thank reviewers for the positive comments and encouraging remarks:

“The authors observe that the commonly used simple structure, such as multi-scale structure, in image denoising violates the J-invariance and thus cannot be used in BSNs.” (**nv4m**)

“The proposed method is interesting and alleviates the constrained architecture design imposed by J-invariance opening up possibilities for new designs of BSNs.”(**nv4m**, **daq5**, **DQiP**,  **jJkN**, **3t6u**)

“The proposed method outperforms prior arts significantly” (**nv4m**, **daq5**, **jJkN**, **3t6u**)

“The submission is technically sound in my opinion and the advantages and limitations of this work are discussed carefully and honestly.” (**nv4m**, **daq5**, **jJkN**, **3t6u**)

“The submission is written with sufficiently clear definitions and formulas. And the organizations are well-designed.” (**nv4m**, **daq5**, **jJkN**, **3t6u**)


We sincerely appreciate your thorough understanding of the method and careful review of the paper. We are truly grateful for your valuable insights and advice.

---

### Decision · Program_Chairs · 2023-09-21

**Decision:**

Accept (poster)

**Comment:**

This paper presents a method for self-supervised image denoising. Specifically, a patch-unshuffle (and shuffle) operation together with a dilated attention block was proposed to achieve the goal. Experimental results on two real noise datasets (SIDD and DND) show the effectiveness of the proposed method over other self-supervised methods. During the review stage, issues like motivation, novelty/technical details, clarification/presentation, and experiments were raised. The authors provided detailed rebuttal responses which address the above concerns. Since all reviewers are positive on this work, it is clear that this paper is above the threshold for publication.